# DreamHuman: Animatable 3D Avatars from Text

**Nikos Kolotouros**   **Thiemo Alldieck**   **Andrei Zanfir**
**Eduard Gabriel Bazavan**   **Mihai Fieraru**   **Cristian Sminchisescu**
Google Research
{kolotouros,alldieck,andreiz,egbazavan,fieraru,sminchisescu}@google.com

## Abstract

We present *DreamHuman*, a method to generate realistic animatable 3D human avatar models solely from textual descriptions. Recent text-to-3D methods have made considerable strides in generation, but are still lacking in important aspects. Control and often spatial resolution remain limited, existing methods produce fixed rather than animated 3D human models, and anthropometric consistency for complex structures like people remains a challenge. *DreamHuman* connects large text-to-image synthesis models, neural radiance fields, and statistical human body models in a novel modeling and optimization framework. This makes it possible to generate dynamic 3D human avatars with high-quality textures and learned, instance-specific, surface deformations. We demonstrate that our method is capable to generate a wide variety of animatable, realistic 3D human models from text. Our 3D models have diverse appearance, clothing, skin tones and body shapes, and significantly outperform both generic text-to-3D approaches and previous text-based 3D avatar generators in visual fidelity.

## 1   Introduction

The remarkable progress in Large Language Models [46, 8] has sparked considerable interest in generating a wide variety of media modalities from text. There has been significant progress in text-to-image [49, 50, 52, 67, 10, 34], text-to-speech [37, 41], text-to-music [2, 19] and text-to-3D [22, 43] generation, to name a few. Key to the success of some of the popular generative image methods conditioned on text has been diffusion models [52, 50, 55]. Recent works have shown these text-to-image models can be combined with differentiable neural 3D scene representations [5] and optimized to generate realistic 3D models solely from textual descriptions [22, 43].

Controllable generation of photorealistic 3D human models has been in the focus of the research community for a long time. This is also the goal of our work; we want to generate realistic, animatable 3D humans given only textual descriptions. Our method goes beyond static text-to-3D generation methods, because we learn a dynamic, articulated 3D model that can be placed in different poses, without additional training or fine-tuning. We capitalize on the recent progress in text-to-3D generation [43], neural radiance fields [31, 5] and human body modelling [64, 3] to produce 3D human models with realistic appearance and high-quality geometry. We achieve this without using any supervised text-to-3D data, or any image conditioning. We generate photorealistic and animatable 3d human models by relying only on text, as can be seen in Figure 1 and Figure 2. As impressive as general-purpose 3D generation methods [43] are, we argue these are suboptimal for 3D human synthesis, due to limited control over generation which often results in undesirable visual artifacts such as unrealistic body proportions, missing limbs, or the wrong number of fingers. Such inconsistencies can be partially attributed to known problems of text-to-image networks, but become even more apparent when considering the arguably more difficult problem of 3D generation. Besides enabling animation capabilities, we show that geometric and kinematic human priors can resolve anthropometric consistency problems in an effective way. Our proposed method, coined

37th Conference on Neural Information Processing Systems (NeurIPS 2023).

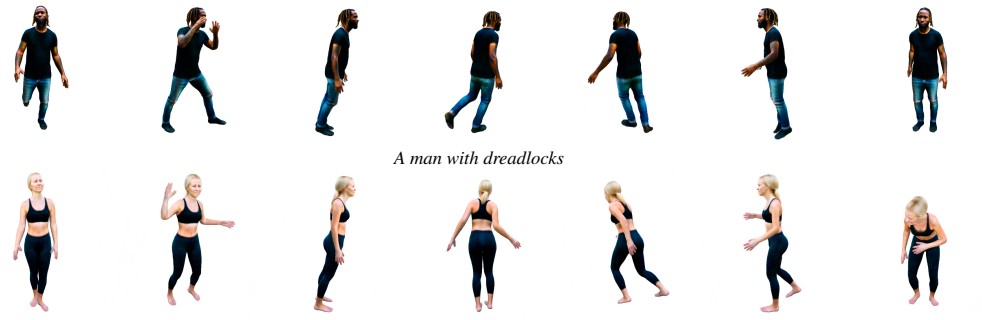

*A man with dreadlocks*

*A blonde woman wearing yoga pants*

Figure 1: **Example of 3D models synthesized and posed by our method**. *DreamHuman* can produce an animatable 3D avatar given only a textual description of a human's appearance. At test time, our avatar can be reposed based on a set of 3D poses or a motion, without additional refinement.

*DreamHuman*, can become a powerful tool for professional artists and 3D animators and can automate complex parts of the design process, with potentially transformative effects in industries such as gaming, special effects, as well as film and content creation.

Our main contributions are:

- We present a novel method to generate 3D human models that can be placed in a variety of poses, with realistic clothing deformations, given only a single textual description, and by training without any supervised text-to-3D data.

- Our models incorporate 3D human body priors that are necessary for regularizing the generation and re-posing of the resulting avatar, by using multiple losses to ensure the quality of human structure, appearance, and deformation.

- We improve the quality of the generation by means of semantic zooming with refining prompts to add detail in perceptually important body regions, such as the face and the hands.

## 2 Related Work

There is considerable work related to diffusion models [58] and their applications to image generation [17, 35, 11, 52, 50, 55, 54] or image editing [24, 53, 16, 32]. Our focus is on text-to-3D [22, 43, 47] and more specifically on realistic 3D human generation conditioned on text prompts. In the following subsections we revisit some of the relevant work related to our goals.

**Text-to-3D generation.** CLIP-Forge [56] combines CLIP [45] text-image embeddings with a learned 3D shape prior to generate 3D objects without any labeled text-to-3D pairs. DreamFields [22] optimizes a NeRF model given a text prompt using guidance from CLIP [45]. CLIP-Mesh [25] also uses CLIP, but substitutes NeRF with meshes as its underlying 3D representation. DreamFusion [43] builds on top of DreamFields and uses supervision from a diffusion-based text-to-image-model [54]. Latent-NeRF [30] uses a similar strategy with DreamFusion, but optimizes a NeRF that operates in the space of a Latent Diffusion model [52]. TEXTure [51] takes as input both a text prompt and a target mesh and optimizes the texture map to agree with the input prompt. Magic3D [28] uses a 2-stage strategy that combines Neural Radiance Fields with meshes for high resolution 3D generation. Unlike our method, all mentioned works produce a static 3D scene given a text prompt. When queried with human related prompts, results often exhibit artifacts like missing face details, unrealistic geometric proportions, partial body generation, or incorrect number of body parts like legs or fingers. We generate accurate and anthropomorphically consistent results by incorporating 3D human priors in the loop.

**Text-to-3D human generation.** Several methods [40, 60, 4, 26, 15] learn to generate 3D human motions from text by leveraging text-to-MoCap datasets. MotionCLIP [59] learns to generate 3D human motions without using any paired text-to-motion data by leveraging CLIP as supervision. However, all these methods output 3D human motions in the form of 3D coordinates or human body model parameters [29] and do not have the capability to generate photorealistic results. AvatarCLIP [18] learns a NeRF in the rest pose of SMPL [29] which is then converted back to a mesh using

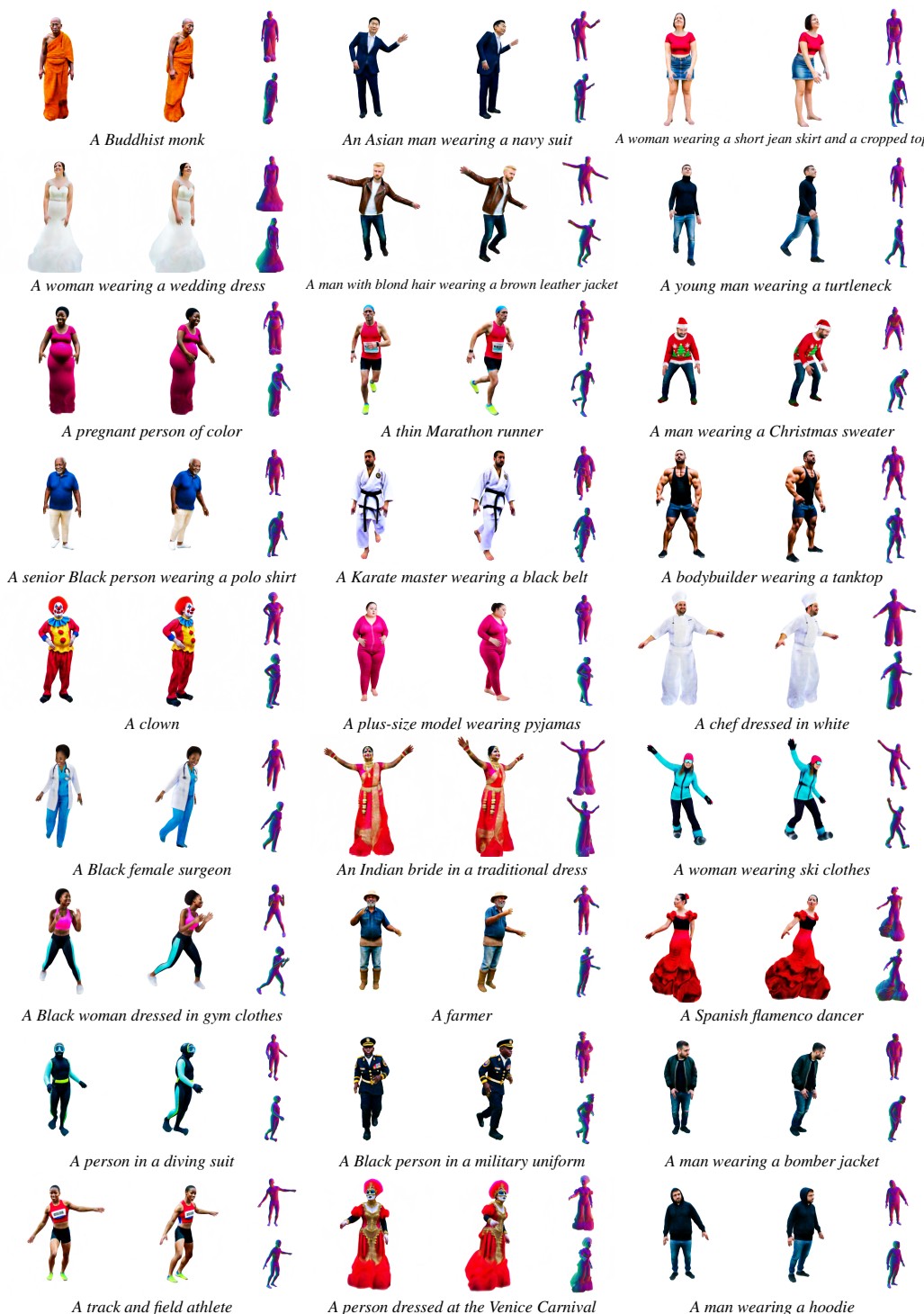

*A Buddhist monk* — *An Asian man wearing a navy suit* — *A woman wearing a short jean skirt and a cropped top*

*A woman wearing a wedding dress* — *A man with blond hair wearing a brown leather jacket* — *A young man wearing a turtleneck*

*A pregnant person of color* — *A thin Marathon runner* — *A man wearing a Christmas sweater*

*A senior Black person wearing a polo shirt* — *A Karate master wearing a black belt* — *A bodybuilder wearing a tanktop*

*A clown* — *A plus-size model wearing pyjamas* — *A chef dressed in white*

*A Black female surgeon* — *An Indian bride in a traditional dress* — *A woman wearing ski clothes*

*A Black woman dressed in gym clothes* — *A farmer* — *A Spanish flamenco dancer*

*A person in a diving suit* — *A Black person in a military uniform* — *A man wearing a bomber jacket*

*A track and field athlete* — *A person dressed at the Venice Carnival* — *A man wearing a hoodie*

Figure 2: **3D human avatars generated using our method given text prompts**. We render each example in a random pose from two viewpoints, along with corresponding surface normal maps.

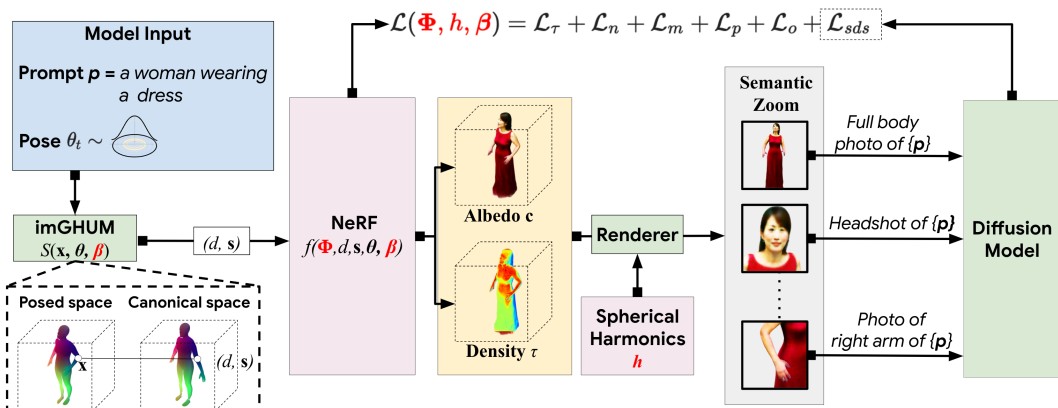

Figure 3: **Overview of *DreamHuman***. Given a text prompt, such as *a woman wearing a dress*, we generate a realistic, animatable 3D avatar whose appearance and body shape match the textual description. A key component in our pipeline is a deformable and pose-conditioned NeRF model learned and constrained using imGHUM [3], an implicit statistical 3D human pose and shape model. At each training step, we synthesize our avatar based on randomly sampled poses and render it from random viewpoints. The optimisation of the avatar structure is guided by the Score Distillation Sampling loss [43] powered by a text-to-image generation model [54]. We rely on imGHUM [3] to add pose control and inject anthropomorphic priors in the avatar optimisation process. We also use several other normal, mask and orientation-based losses in order to ensure coherent synthesis. NeRF, body shape, and spherical harmonics illumination parameters (in red) are optimised.

marching cubes. However the reposing procedure depends on fixed skinning weights that limit the overall realism of the animation. In contrast, our method learns per-instance pose-specific geometric deformations that result in significantly more realistic clothing appearance. Concurrent work AvatarCraft [23] produces avatars with geometry very much tied to the underlying SMPL [29] model and thus cannot model loose-fitting clothing. Another concurrent work, DreamAvatar [9], suffers from the fact that it to be retrained every time for a new pose, which makes it computationally prohibitive to repose.

**Deformable Neural Radiance Fields.** Several methods attempt to learn Deformable NeRFs to model dynamic content [38, 44, 61, 39, 57]. There has also been work on representing articulated human bodies [65, 36, 63, 69, 20, 66, 27]. The method more closely related to ours is H-NeRF [65], which combines implicit human body models with NeRFs. Compared to H-NeRF, our method uses a simpler approach where we enforce consistency directly in 3D and not via renderings of two different density fields. Also, while H-NeRF that uses videos for supervision, our only input is text, and we use are not constrained by the poses and viewpoints present on the video. Thus our method can generalize better in a variety of different poses and camera viewpoints.

## 3 Methodology

### 3.1 Architecture

We rely on Neural Radiance Fields (NeRF) [31] to represent our 3D scene as a continuous function of its spatial coordinates [33]. We use a multi-layer perceptron (MLP) that maps each spatial point $\mathbf{x} \in \mathbb{R}^3$ to a tuple $(\mathbf{c}, \tau)$ of RGB color and density values. To render a scene using NeRF, one needs to cast rays from the camera center passing through the image pixels and then compute the expected color $\mathbf{C}$ along each ray. In practice, this is done by sampling points $\mathbf{x}_i$ on the ray and then approximating the rendering integral [5]

$$\mathbf{C} = \sum_i w_i \mathbf{c}_i, \quad w_i = \alpha_i \prod_{j<i}(1 - \alpha_j), \quad \alpha_i = 1 - \exp\left(-\tau_i \|\mathbf{x}_{i+1} - \mathbf{x}_i\|\right). \quad (1)$$

While NeRF provides a general purpose scene representation, we aim to regularize the optimised geometry and appearance using human structural priors. To that effect, we use imGHUM [3], which is the implicit version of the GHUM [64] body model, and thus compatible with neural scene representations. Given pose $\boldsymbol{\theta}$ and shape $\boldsymbol{\beta}$ parameters, imGHUM predicts a semantic signed distance

function $S(\mathbf{x}, \boldsymbol{\theta}, \boldsymbol{\beta})$ that maps a 3D spatial point $\mathbf{x}$ to a tuple $(d, \mathbf{s})$ containing the signed distance $d$ of the point from the body surface together with a semantic correspondence code $\mathbf{s} \in \mathbb{R}^3$ that associates $\mathbf{x}$ with the nearest surface point on the body.

Our model architecture uses mip-NeRF 360 [5] for the NeRF backbone. An overview can be seen in Figure 3. Specifically, we modify each of the MLPs in the standard NeRF model in order to operate in the imGHUM [3] semantic signed distance space instead of the standard 3D coordinates. Given a 3D point $\mathbf{x} \in \mathbb{R}^3$ and pose and shape parameters $\boldsymbol{\theta}$ and $\boldsymbol{\beta}$ respectively, we first encode it with imGHUM [3] into the 4D semantic descriptor $(d, \mathbf{s}) = S(\mathbf{x}, \boldsymbol{\theta}, \boldsymbol{\beta})$. We can then learn a NeRF $f$ in this semantic space

$$(\mathbf{c}, \tau) = f(\boldsymbol{\Phi}, d, \mathbf{s}). \tag{2}$$

where $\boldsymbol{\Phi}$ represents the trainable weights for the NeRF module. Similarly with DreamFusion [43], $\mathbf{c}$ models the albedo of the surface at the corresponding point, and we use this together with the learnt geometry to produce shaded renderings.

By learning a NeRF in the semantic signed distance space of a human body model, we learn a representation that can generalize to different human poses and body shapes. This is because the local geometry and color are generally preserved in $(d, \mathbf{s})$ for different shape and pose parameters. One can think of the process similarly to learning a NeRF for the template pose and then warping to new shapes and poses by leveraging the 3D correspondences from the body model [42, 6]. However, animating the model in different poses is challenging. Clothing deformations work reasonably only for tight-fitting clothing or for accessories that are usually moving rigidly with the body, such as hats and glasses. For this reason, we propose to augment and modulate the NeRF input with pose and shape parameters, thus giving it the capability to model non-rigid pose and surface dependent effects beyond the body shape itself. By doing so, the model can learn per-instance, pose-dependent deformations of the clothing surface, on top of what the imGHUM model can represent. Thus, our NeRF input becomes

$$(\mathbf{c}, \tau) = f(\boldsymbol{\Phi}, d, \mathbf{s}, \boldsymbol{\theta}, \boldsymbol{\beta}). \tag{3}$$

To make sure the NeRF model conforms to the underlying body geometry we propose to calculate the final density as the maximum of the density $\tau$ computed by the NeRF MLP and the density proxy $\hat{\tau}(d) = a\sigma(-ad)$ computed from imGHUM based on the signed distance value. In the previous equation, $\sigma$ is the sigmoid function and $a$ a positive constant that controls the sharpness of the density field. Effectively $\hat{\tau}$ is a smooth scaled indicator function, with $\hat{\tau} \approx \alpha$ inside the body and $\hat{\tau} \approx 0$ outside. In this way we avoid undesirable artifacts, such as the model removing limbs or fine structure like fingers, unless the prompt indicates so.

**Shading and rendering model.** We found that a diffuse reflectance model [43] does not produce very realistic renderings of the human appearance, with results that often look cartoon-like. Hence we rely on a spherical harmonics lighting model [48] and preserve the first 9 components. During NeRF training, we additionally optimize for the spherical harmonics coefficients (i.e. $\mathbf{h} \in \mathbb{R}^{1 \times 10}$). However, by using just the optimized coefficients can lead to inadequate albedo-shading disentanglement and occasionally some geometric regions may never get highlights. Empirically, we found that sampling random coefficients a fraction of the time during training produces better results.

**Semantic zoom.** One limitation in using a text-to-image diffusion model for supervision is its $64 \times 64$ pixels input resolution. As a result textures are often blurry and the geometry lacks fine details. One way around this would be the use of super-resolution diffusion models, e.g. the $64 \times 64 \rightarrow 256 \times 256$. However these make rendering very expensive as memory requirements increase by a factor of 16. By using a human body model with attached semantics like imGHUM to control the NeRF, we benefit from direct correspondences between the 3D space occupancy and the human body parts. We can then very easily infer the location of important body parts such as the head, hands, etc. for any given pose. Therefore, during optimization we propose to use this information to zoom in on different parts of the body, thus increasing the effective model resolution. This can leverage both detail implicit in the image diffusion model used, and structure in the imGHUM human body prior. Instead of rendering a $64 \times 64$ image of the whole body, we render instead a $64 \times 64$ image of the head and some body parts where fine details are important. In total, we define 6 semantic regions: head, upper body, lower body, midsection, left arm, right arm. We also modify the text prompt accordingly, in order to explicitly encode this information in the text. In contrast to AvatarCLIP that only zooms-in on the face, zooming-in on all body parts results in much crisper textures and geometric detail throughout. For more information please check our Supplementary Material.

## 3.2 Loss functions

**imGHUM density loss.** To enforce that the estimated avatar follows the underlying body shape geometry, we add an $L_1$ loss between the NeRF density and a density proxy computed from imGHUM. This loss encourages sparse modifications in the body geometry and is necessary to preserve important geometric details on the body. The density loss is defined as

$$\mathcal{L}_\tau = ||\tau - \hat{\tau}||_1 . \tag{4}$$

**Predicted normal loss.** Following Ref-NeRF [62], we modify the MLP to also predict the surface normal vector $\mathbf{n}'$ at each spatial location and then add a loss between the predicted normals and the normals $\mathbf{n}$ obtained from the gradient of the density field. The normal loss is

$$\mathcal{L}_n = \sum_i w_i \, ||\mathbf{n}' - \mathbf{n}|| . \tag{5}$$

In our case, this loss serves two purposes: it acts as a smoothness loss on the surface normals and also helps learning the pose-dependent deformations. Regarding the first part, we noticed that for clothing such as skirts or dresses with uniform dark texture, the resulting surface normals are often very noisy, resulting in sub-optimal shading results. Naturally, the predicted surface normals are smoother that the density normals because of the spectral bias of MLPs and hence this loss acts as a surface regularizer. More importantly though, the auxiliary task of predicting the surface normals encourages the MLP to use the pose conditioning information during optimization. The pose-dependent density deformations are sparse and subtle since a considerable part of the work is usually handled decently by imGHUM. Hence, it is easy for the MLP to ignore the conditioning on the pose parameters because it has a small overall impact on the loss. Note, however, that pose conditioning is necessary in order to predict the correct surface normals. If not used, then the predicted normal vector at a particular point on the surface, e.g. on the arm, will be always the same, regardless of the limb orientation, because it only depends on the canonical coordinates $(d, \mathbf{s})$.

**Foreground mask loss.** The above density loss forces the NeRF to respect the underlying body geometry and disentangles the subject from the background. However, we noticed that in some cases this can result in making the clothing or hair translucent. To prevent it, we add a loss on the rendered mask $M$ that encourages it to be binary. The loss is defined as

$$\mathcal{L}_m = \frac{1}{HW} \sum_{x=1}^{H} \sum_{y=1}^{W} \min \left( \log M(x,y), \log(1 - M(x,y)) \right) \tag{6}$$

where $M(x,y) = \sum_i w_i$, i.e. the sum of the rendering weights for the ray through pixel $(x,y)$.

**Diffusion Models and Score Distillation Sampling.** Diffusion models are a class of generative models that learn to produce samples from a target distribution by iteratively denoising samples coming from a tractable base distribution. They consist of a fixed forward process that gradually transforms a sample $\mathbf{u}$ from the data distribution to Gaussian noise and a learnable reverse process that approximates the inverse of the forward process.

To generate images from the data distribution given an NeRF with parameters $\mathbf{\Phi}$, [43] proposed to use Score Distillation Sampling. This involves optimizing an approximation of the diffusion model training loss. The gradient of the Score Distillation Sampling loss with respect to the NeRF is defined as

$$\nabla \mathcal{L}_{sds} = \mathbb{E}_{t \sim \mathcal{U}[0,1], \epsilon \sim \mathcal{N}(\mathbf{0}, \mathbf{I})} \left[ w_s(t) \left( \hat{\epsilon}(\mathbf{z}_t; y, t) - \epsilon \right) \frac{\partial \mathbf{u}}{\partial \mathbf{\Phi}} \right] . \tag{7}$$

where $\epsilon$ is the injected noise, $z_t$ the noisy rendered image and $\hat{epsilon}$ the noise prediction from the diffusion model. We use the SDS loss [43, 54] to supervise the 3D generation given the actively modified semantic-zoom prompts.

**Additional losses.** We use the orientation loss $\mathcal{L}_o$ from Ref-NeRF [62] that penalizes 'back-facing' normals for points along the ray that are visible, as well as the loss on the proposal weights $\mathcal{L}_p$ in mip-NeRF360 [5].

Our full loss function then becomes

$$\mathcal{L} = \mathcal{L}_{sds} + \mathcal{L}_o + \mathcal{L}_p + \mathcal{L}_m + \mathcal{L}_n + \mathcal{L}_\tau \tag{8}$$

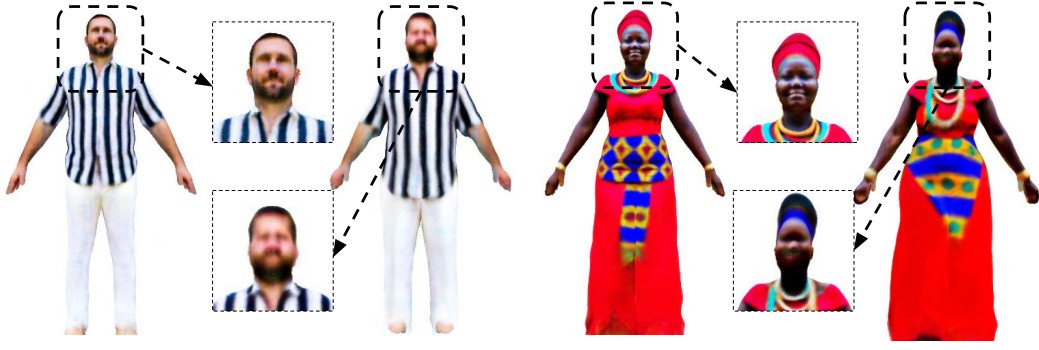

*A man wearing a striped shirt and white linen pants*   *An African woman dressed in traditional clothes*

Figure 4: **Importance of semantic zoom**. For each example, the left image shows the generated avatar with semantic zoom, whereas the right image an avatars generated without it. Notice how the semantic zoom allows us to reconstruct sharper, higher-quality textures.

## 3.3 Optimization

**Body pose sampling.** Previous methods like H-NeRF [65] and Human-NeRF [63] have limited generalization capabilities because they are only trained on poses and viewpoints that are present in an input video. Our method on the other hand does not have such constraints. At each optimization step, we sample a random pose from a distribution [68] trained on 3D motion capture [1, 21, 12, 14, 13] and use this to pose imGHUM. Sampling different poses is necessary for learning the dependency of the surface geometry on the model shape and pose parameters. At the same time it helps disentangle the generated avatar from objects in the background. Without the pose randomization strategy often times there is not sufficient disentanglement of the avatar geometry from the background and the final geometry includes additional objects such as the ground floor, or even the shadow of the person around the legs

**Other details.** We optimize the NeRF and the imGHUM shape parameters $\beta$ instead of randomly sampling shape parameters. This is because the body shape is often explicitly or implicitly described in the caption. We generate one avatar with an underlying body shape given all constraints coming from the text prompt and the related losses. Similarly with DreamFusion, we randomly sample camera positions in spherical coordinates and then augment the input prompt with view-dependent conditioning based on the azimuth and elevation. We also randomly select the radius $r$ from the origin as well as the focal length of the camera. For additional details please see our Supplementary Material.

## 4 Experiments

In this section we illustrate the effectiveness of our proposed method. We show how the individual proposed components help, and how we compare to recent state-of-the-art methods. Figure 2 shows a wide variety of generated 3d human models in different poses, so we can illustrate diverse body shapes, skin tones and body compositions. Due to space constraints, additional results are available in the Supplementary Material.

### 4.1 Ablation Study

**Semantic zoom.** In Figure 4, we show the importance of our semantic zoom strategy. Notice how our method is able to generate much higher-quality textures, both for the body and the face.

**Pose-dependent deformations.** In Figure 5, we show examples of how we can learn realistic garment deformations. In the example of the ballerina, one can see that the skirt deforms more naturally when the legs move. On the other hand the baseline without non-rigid deformations struggles to capture the skirt geometry and exhibits floating artifacts around the legs. Similar observations can be made for the man wearing shorts. We hypothesize that our model can infer this because the text-to-image generator has been trained on lots of images of people wearing clothes in different poses. Therefore,

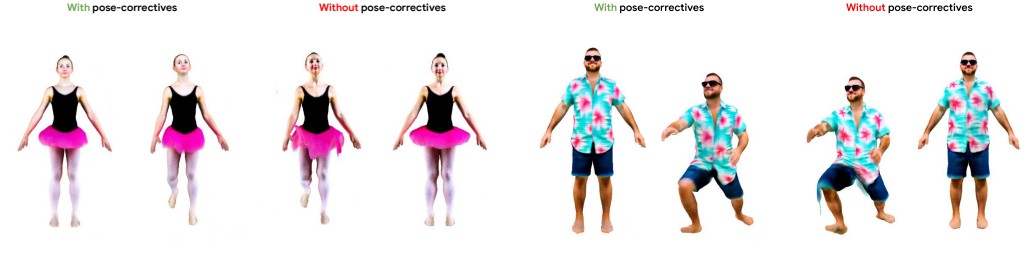

*A ballerina*       *A man wearing a Hawaiian shirt, sunglasses and shorts*

Figure 5: **Importance of pose-dependent deformations and pose sampling in the NeRF model,** $f(\mathbf{\Phi}, d, \mathbf{s}, \boldsymbol{\theta}, \boldsymbol{\beta})$**.** Our non-rigid pose-dependent deformations enable more realistic clothing when reposing the avatar. For each of the two example prompts we show two generated avatars, with and without pose-correctives. Notice how the skirt and the shorts move more naturally when reposing the avatar.

our model, although not using video, or relying on a text-to-video diffusion loss, can leverage general knowledge on how clothing drapes.

**Choice of diffusion model.** Our method is not tied to a particular diffusion model. To demonstrate this, we substitute Imagen [55] with the open-source Stable Diffusion model [52]. Stable Diffusion is a latent diffusion model, so at each training iteration we use our NeRF to render an RGB image, pass it through the latent encoder, and then apply the SDS loss on the latent embeddings. In Figure 6, we show qualitative results of our method trained using Stable Diffusion. Overall we observe that we are able to generate high quality avatars, regardless of the choice of the diffusion model.

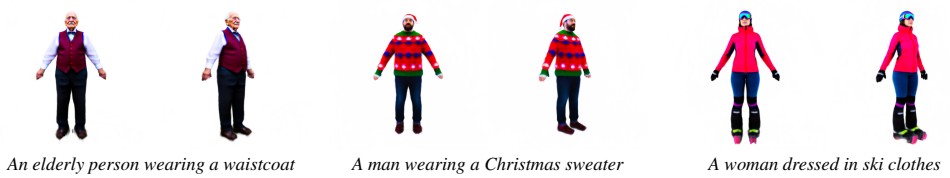

*An elderly person wearing a waistcoat*    *A man wearing a Christmas sweater*    *A woman dressed in ski clothes*

Figure 6: Generation results using Stable Diffusion.

## 4.2 Comparison with the state of the art

**Qualitative Evaluations.** In Figure 7, we show a qualitative comparison of our method with DreamFusion. DreamFusion suffers from limited control over generation. Even though it was prompted to generate the full body of the subject, very often it produces a 3D model of the upper body, or the head. At the same time it cannot properly disentangle the human subject from other objects in the scene, resulting in 3D models that contain parts of the environment. More importantly though, it very often produces unpleasant visual artifacts, such as non-realistic body proportions, missing or multiple limbs, as well as degenerate geometry that can be attributed to viewpoint overfitting. Our method is able to overcome these issues by utilizing a strong 3D prior on human body geometry. For more comparisons we refer the readers to the Supplementary Material.

Figure 8 shows a comparison between *DreamHuman* and AvatarCLIP. We can see that our method is able to generate significantly better geometry and texture quality. The geometry of the reconstructed avatars with AvatarCLIP is very close to the underlying body model geometry, with only minor modifications. As a result, it cannot handle loose-fitting clothing, dresses, and accessories like hats. The model textures from AvatarCLIP also have significant artifacts and do not match the realism and overall quality of *DreamHuman* in all examples we tried.

**Quantitative evaluation using CLIP.** Following common practice, we also use CLIP to evaluate the alignment of the rendered 3D models with the input text prompts. We use a total of 160 prompts with descriptions of people. The results are shown in Table 1. We can see that our method consistently outperforms DreamFusion. We also include comparisons with AvatarCLIP. However, note that AvatarCLIP is trained using CLIP as supervision, meaning that the CLIP-based metrics are biased

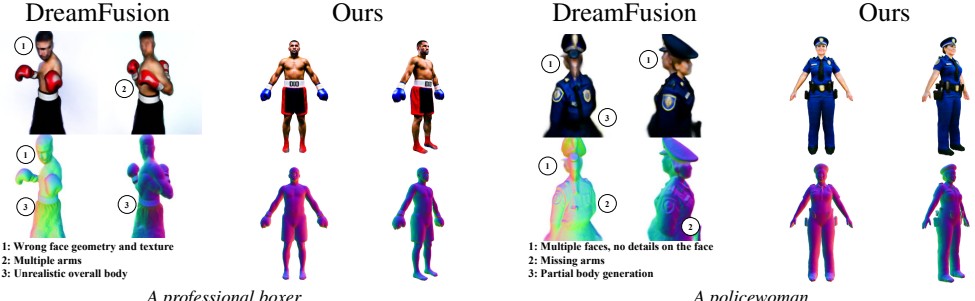

1: Wrong face geometry and texture
2: Multiple arms
3: Unrealistic overall body

1: Multiple faces, no details on the face
2: Missing arms
3: Partial body generation

*A professional boxer*                              *A policewoman*

Figure 7: **Comparison with DreamFusion**. For each example we show the rendered 3D model as well as the corresponding surface normals. Both methods were asked to reconstruct the full body of the subject by prepending *A DSLR full body photo* to the prompt.

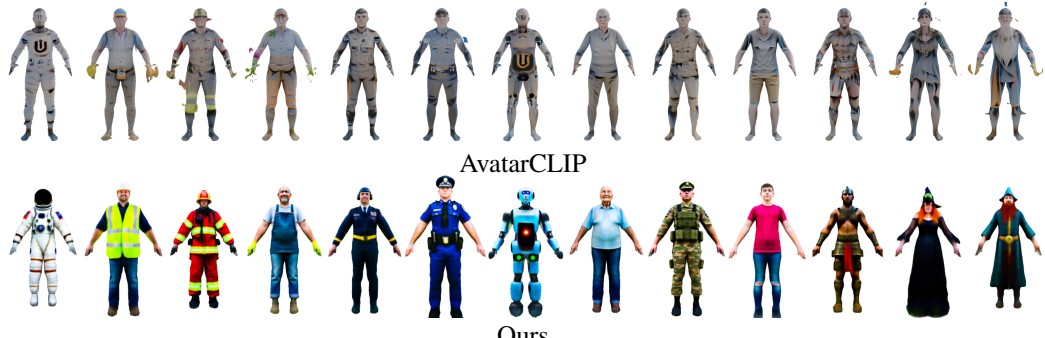

AvatarCLIP

Ours

Figure 8: **Comparison with AvatarCLIP**. We compare *DreamHuman* with AvatarCLIP [18]. From left to right we used the following prompts: *astronaut, construction manager, firefighter, gardener, pilot, police officer, robot, senior citizen, soldier, teenager, warrior, witch, wizard*. Notice that our method generates much more realistic texture and geometry. All illustrations are in default A-Pose.

favorably towards AvatarCLIP. This means that a direct comparison to other methods is potentially unfair.

Table 1: **Evaluation of the rendered 3D models using CLIP**. We report the R-Precision as well as whether the true caption is in the top 3 and 5 highest-scoring captions.

| Method | R-Precision ↑ | Top-3 ↑ | Top-5 ↑ |
|---|---|---|---|
| DreamFusion [43] | 0.775 | 0.888 | 0.925 |
| AvatarCLIP [18] | 0.855 | 0.962 | 0.981 |
| Ours | 0.838 | 0.931 | 0.956 |

**User study.** We conducted a user study to assess the quality of our 3D avatar generation pipeline. We used 20 text prompts from the AvatarCLIP website, selected from the *General Description* category. We ran our method on those 20 text prompts and rendered the final results in the rest pose, from the front and from the side. We did the same for the precomputed meshes that the AvatarCLIP authors provide on their website. We then asked the users to rate the two methods on (a) the perceived agreement between the renderings and the input text, and (b) the perceived visual quality of the generated avatar. The ratings were on a scale from 1-5, with 1 meaning *Very Bad* and 5 *Very Good*. The results in Table 2 show that the raters consistently preferred our method over AvatarCLIP.

### 4.3   Generation variability.

Here we investigate whether there is variation in our generations for the same prompt, but with different random seeds. As originally observed in DreamFusion [43], there is limited variation in the generations, probably because the SDS loss is mode-seeking, so it tends to latch on to specific modes of the distribution. Our results demonstrate some level of variability, as seen on Figure 9.

Table 2: **User study on 3D avatar generation quality.**

| Method | Agreement with text ↑ | Visual quality ↑ |
|---|---|---|
| AvatarCLIP [18] | 2.96 | 2.49 |
| Ours | **4.45** | **4.16** |

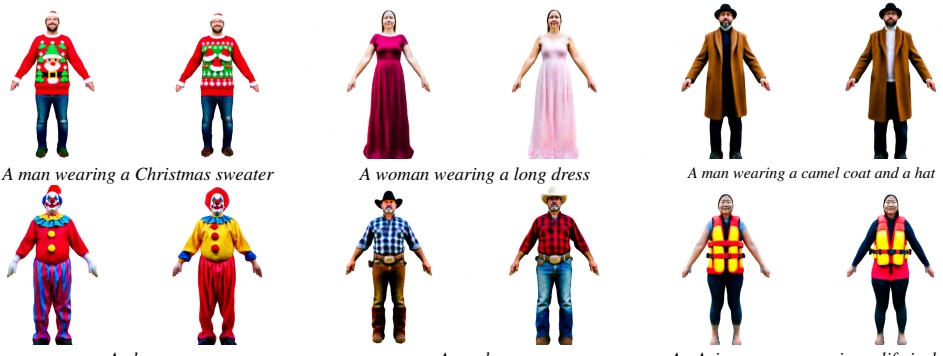

A man wearing a Christmas sweater       A woman wearing a long dress       A man wearing a camel coat and a hat

A clown       A cowboy       An Asian woman wearing a life jacket

Figure 9: **Generations using different random seeds.** For each example we show the generated 3D avatar using 2 different random seeds as initialization.

## 5   Conclusion

We presented *DreamHuman*, a novel method for generating 3D human avatars from text. Our method leverages statistical 3D human body models and recent advances in 3D modelling and text-to-3D generation to create animatable 3D human avatars, without any paired text-to-3D supervision. We illustrated that our method can generate photorealistic results, with detailed geometry and outperforms the state of the art by a large margin.

**Limitations and future work.** Since our model is trained without any 3D data, it sometimes draws fine details like wrinkles using the albedo map instead of creating them based on geometry. Future work can address this by leveraging 3D data to resolve some of the reconstruction ambiguities. Additionally, the model sometimes cannot properly disentangle albedo from shading, resulting in baked reflections and shadows. Current computational constraints from the diffusion models prevent us from scaling the method to very high resolution textures and geometric detail like hair. Finally, the realism of clothing animation can benefit from a video model.

**Broader Impact.** While our method does not use any additional training data, it relies in part on text-to-image diffusion models which have been pre-trained on large-scale datasets containing sometimes insufficiently curated images and captions [7] (N.B. the level of effective automation to guarantee the removal of undesired content is considerable for most models we use in this paper). Also, text-to-image generators use LLMs for the text encoder, pre-trained on uncurated internet-scale data. Although our method uses statistical 3D human body shape models learnt using highly curated and diverse data to remove bias, ultimately our generation process may be vulnerable to some bias in its dependencies.

The goal of our method is to generate 3D models of people, which has the potential to be misused in connection with fake media. However, it is important to highlight that our rendered 3D human models are typically less realistic than their 2D-generated counterparts. Regardless, in practical settings, safeguards should be used to prevent abuse, such as filtering the input text prompts and detecting any unsafe content in the model renderings.

Our method has the ability to generate people with diverse body shapes, appearance, skin color and clothing. This can enable the generation of diverse large-scale synthetic 3D datasets for human-related tasks, and in turn may support training models with fairer outcomes across different groups.

*DreamHuman* can potentially augment the work of artists and other creative professionals. It could be used as a complementary tool to boost productivity. It also has the potential to democratize 3D content creation that currently requires specialized knowledge and expensive proprietary software.

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
