# *Supplementary Material* for DreamHuman: Animatable 3D Avatars from Text

This document contains additional details and experiments that did not fit in the main text due to space constraints. For animations and additional results please also check the included videos.

## 1 Model and training Details

We use a similar optimization strategy with DreamFusion, so unless otherwise noted the hyperparameters remain the same. For example, we use the Distributed Shampoo optimizer [2]. Similarly with DreamFusion we also train on a TPUv4 machine with 4 chips. That gives us an effective batch size of 4 renderings per training step. The main changes in the optimization procedure are the following:

- We increase the number of optimization iterations from 15,000 to 50,000. This was necessary to give the network more time to learn the pose-dependent deformations, since we train with multiple poses. We did not observe any significant benefits by training for more iterations.

- For the first 3000 iterations we only pose the model in the rest pose. This helps with faster learning of the initial rough geometry and texture because the model does not get distracted by the pose variations.

- After the first 3,000 iterations, at each optimization step we sample 4 different camera and pose configurations and use those to render our model in parallel. At the same time the rest pose has on average less self occlusions in 3D compared to random poses and this is also helpful in the start of training.

- We also enable the semantic zooming strategy after 3,000 iterations. This was done because the model textures and geometry are very coarse in the early stages of the optimization, so trying to add details from the beginning is not very meaningful.

- We use a total of 6 semantic regions for zooming. We randomly choose between each of these with a fixed probability, while also manipulate the text prompt accordingly. The strategy is described on Table 1.

- We use a standard positional encoding strategy for our MLP inputs. More specifically we apply the encoding on all 4 dimensions of $(\mathbf{s}, d)$. We found empirically that the color part of our network benefits from the presence of higher frequencies, however this results in a very noisy geometry. For this reason we use a different maximum frequency for the color and density subnetworks ($2^{10}$ and $2^5$ respectively).

## 2 Additional Results

**Body shape manipulation.** One of the design choices that we made is that we jointly optimize the NeRF [3] and the shape parameters of imGHUM [1]. This was chosen because the prompt often describes the body shape of the person, either explicitly or implicitly. For example, a weightlifter has a different physique from a Marathon runner, due to the different requirements of each sport. Even though we didn't train by sampling body shape parameters like we do with pose, we can leverage the canonical mapping of imGHUM and still get plausible generations if we modify the body shape parameters. The reshaping results are shown on Figure 1.

Preprint. Under review.

Table 1: Semantic Zoom Strategy

| Semantic Region | Modified prompt | Selection probability |
|---|---|---|
| full body | *A DSLR full body photo of* {p} | 0.4 |
| face | *A DSLR headshot of* {p} | 0.25 |
| upper body | *A DSLR upper body photo of* {p} | 0.1 |
| lower body | *A DSLR lower body photo of* {p} | 0.1 |
| midsection | *A DSLR photo of the midsection of* {p} | 0.05 |
| left arm | *A DSLR photo of the left arm of* {p} | 0.05 |
| right arm | *A DSLR photo of the right arm of* {p} | 0.05 |

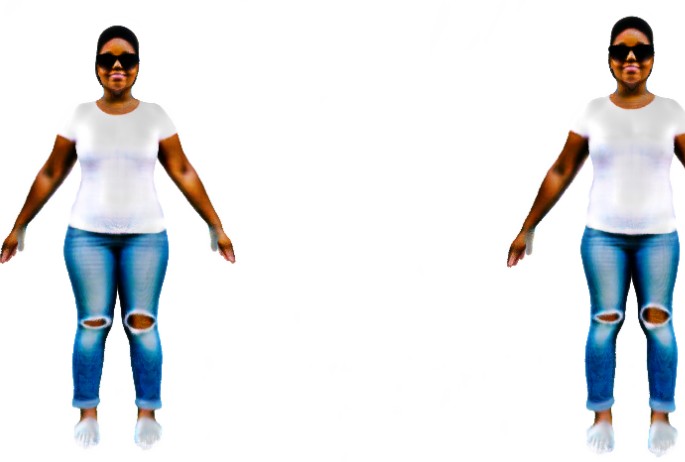

*A Black woman wearing sunglasses, a white t-shirt and jeans*

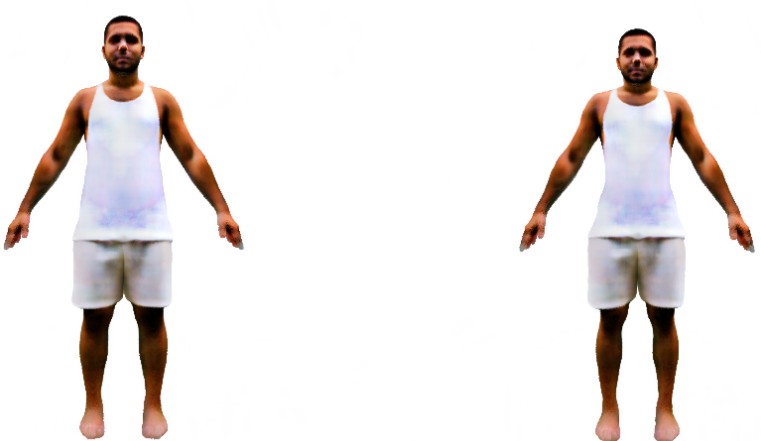

*A man wearing a white tanktop and shorts*

Figure 1: **Reshaping examples**. For each example we show generations using 2 different body shapes. The first example is generated using the optimized imGHUM shape parameters, whereas for the second example we randomly sample shape parameters from the prior distribution.

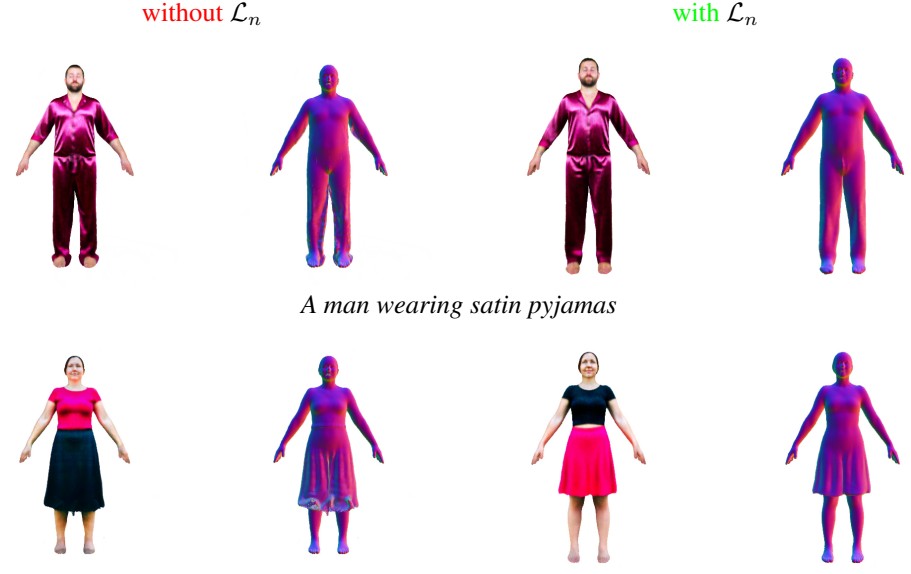

*A man wearing satin pyjamas*

*A woman wearing a skirt*

Figure 2: **Ablation study on** $\mathcal{L}_n$. For each example we show the generated avatar with and without the loss on the predicted normals. Notice how when we do not use $\mathcal{L}_n$ the clothing geometry is very noisy.

**Importance of** $\mathcal{L}_n$**.**  As we mentioned in the main paper, very often the generated surface tends to be noisy, and by adding $\mathcal{L}_n$ we manage to make the surface smoother. Figure 2 shows a comparison between using and not using the normal smoothness loss and demonstrates the effectiveness of $\mathcal{L}_n$ in producing higher-quality surfaces.

**Importance of density bias term.**  As discussed following Eq. (3) of the main paper, the final density is computed as:

$$\tau'(\mathbf{\Phi}, d, \mathbf{s}, \boldsymbol{\theta}, \boldsymbol{\beta}) = \max\left\{\tau(\mathbf{\Phi}, d, \mathbf{s}, \boldsymbol{\theta}, \boldsymbol{\beta}), \hat{\tau}(d)\right\} \tag{1}$$

where $\hat{\tau}(d) = a\sigma(-ad)$ is the density proxy computed from imGHUM based on the signed distance value and $\tau$ the density computed from the MLP. If we do not add this strong bias towards the imGHUM body geometry, then we get artifacts such as removal of limbs or lack of detail in the fingers. Although $\mathcal{L}_\tau$ guides the generation of an avatar that respects the underlying body geometry, fingers or arms are much finer structures compared to the rest of the body. Thus it is relatively cheap to alter their geometry and introduce unpleasant visual artifacts while still achieving a low density loss value. Figure 3 shows a comparison between using and not using the density bias term (1).

**Importance of** $\mathcal{L}_\tau$**.**  Here we evaluate the importance of using the density regularizer $\mathcal{L}_\tau$, as defined on Eq (4) of the main paper. Conceptually $\mathcal{L}_\tau$ discourages the model from significantly altering the underlying body geometry. Unlike the baseline that is trained without $\mathcal{L}_\tau$, our model preserves important details such as facial features. The comparison is shown on Figure 4.

**Importance of spherical harmonics lighting.**  Here we show a qualitative comparison between the shading model of DreamFusion [4] and our shading model that is based on spherical harmonics. The results are shown on Figure 5. By optimizing the Spherical Harmonics coefficients, our method is able to render a scene that looks more photorealistic compared to the vanilla lighting model used in DreamFusion.

**Failure cases.**  Figure 6 shows some failure cases of our method. Some failures can be for the most part attributed to the underlying text-to-image model [5] and the SDS loss [4] that we use. In the first example it fails to generate a checkerboard pattern on the robe. For the man wearing a scarf, part of the scarf is impainted on the body. In the case of the man wearing a kilt, part of the checkerboard texture is also imprinted on the geometry. Last, for extreme poses that are not physically plausible given the affordances of the particular garment, our method can produce incorrect geometry.

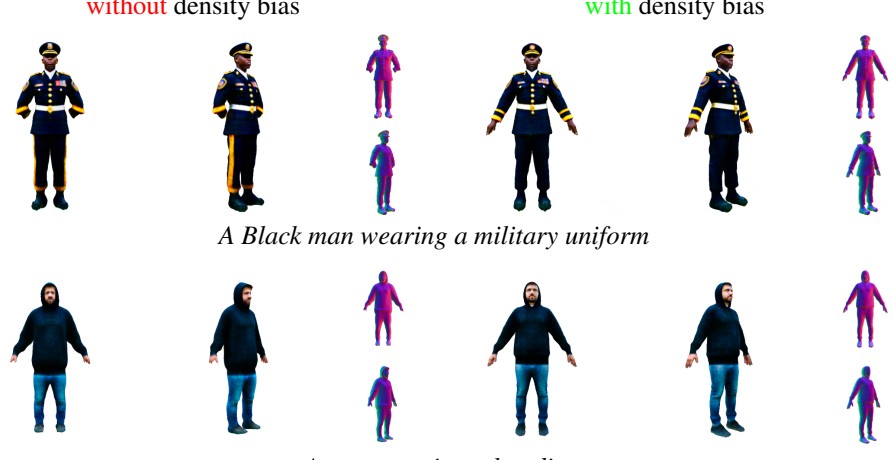

Figure 3: **Ablation study on using the density bias**. For each example we show the generated avatar with and without the imGHUM density bias term. Notice that when we omit the density bias term, it results in the removal of the arms in the first example and the fingers in the second.

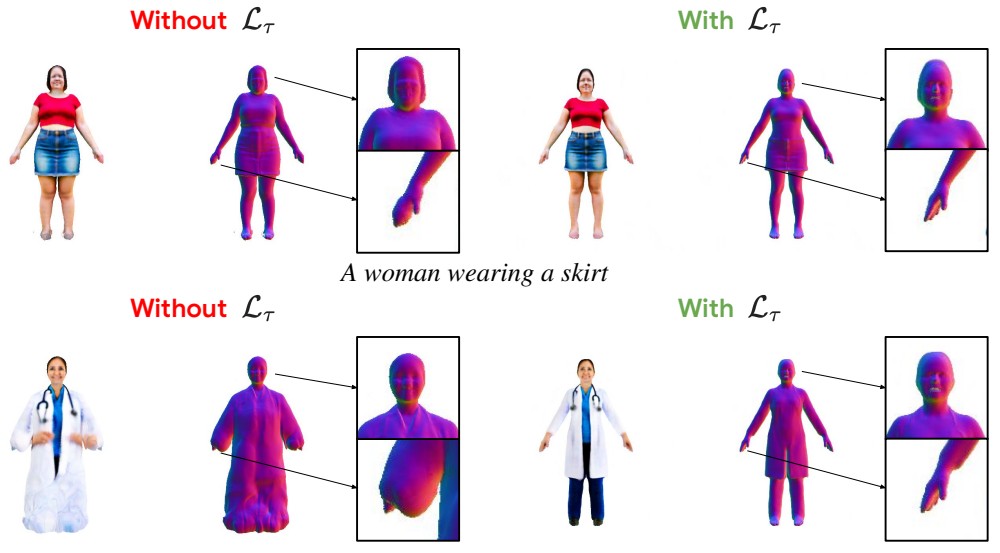

Figure 4: **Ablation study on** $\mathcal{L}_\tau$. For each example we show the generated avatar with and without the density regularizer. Notice the loss of important geometric details on the face and the hands when we don't use it. $\mathcal{L}_\tau$.

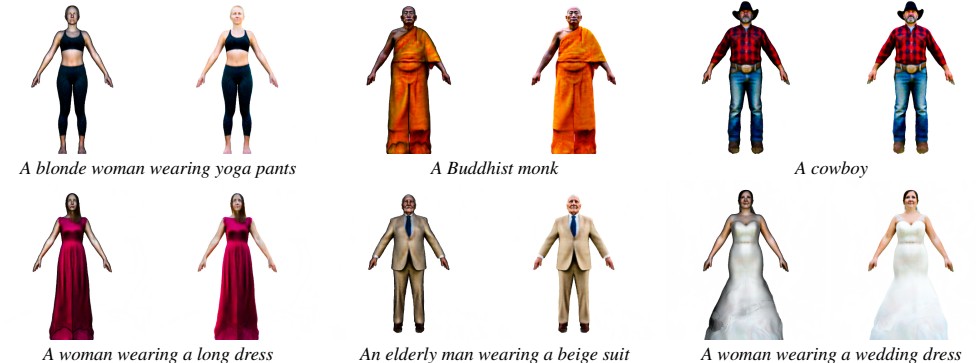

Figure 5: **Ablation study on the use of Spherical Harmonics lighting.** For each example we show the result with and without our learned spherical harmonics lighting. We can see that our model produces higher quality renderings that are more photorealistic.

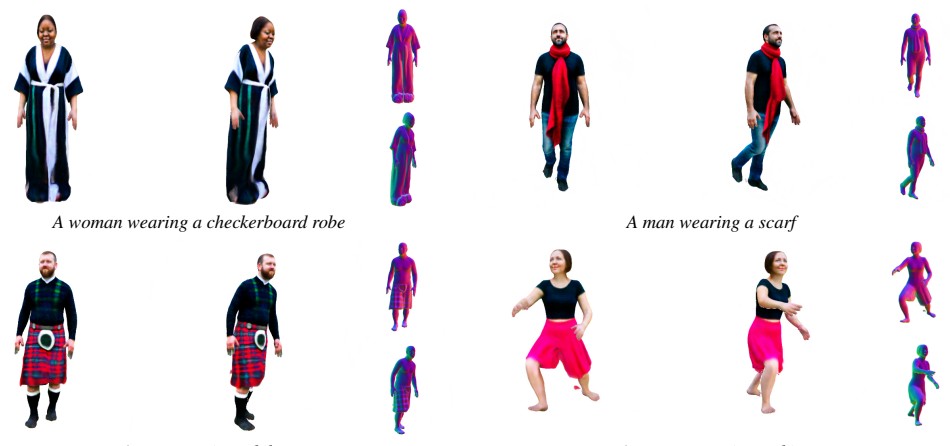

*A woman wearing a checkerboard robe*    *A man wearing a scarf*

*A man wearing a kilt*    *A woman wearing a skirt*

Figure 6: **Failure cases**. We show some example failure cases of our method. Please refer to the text for a more detailed explanation.