# OpenReview forum: "DreamHuman: Animatable 3D Avatars from Text"
_NeurIPS.cc/2023/Conference — NeurIPS 2023 spotlight_

### Official Review · Reviewer_FnWG · 2023-06-27

**Soundness:** 2 fair
**Presentation:** 3 good
**Contribution:** 3 good
**Rating:** 6
**Confidence:** 4

**Summary:**

This is a paper focusing on generating 3D animatable full-body human avatar from text using pretrained 2D diffusion model and Score Distillation Sampling (SDS).
The proposed approach differs from the existing approach in that, instead of directly representing a canonical space using surface template e.g. SMPL, it 1) uses the implicit 3D human model to establish the correspondence and condition the canonical representation on the pose parameter, 2) adopts a per-part optimization, and 3) uses a physics based shading formulation and jointly optimizes the environment lighting. These three technical novelty intent to achieve more plausible deformation for loose clothing, better detail reconstruction for faces and hands, and more realistic colors, respectively.
The comparisons with AvatarClip and DreamFusion demonstrate the advantages of the proposed method.

**Strengths:**

- substantially better visual quality compared to AvatarClip and DreamFusion, although the former only with limited evidence.

- the use of imGHUM seems to improve the diversity of clothing.

- the part-based optimization visibly improves the visual quality of the generation.


**Weaknesses:**

- Even though the following papers can be considered concurrent, they should be mentioned in the related work.
Cao, Yukang, et al. "Dreamavatar: Text-and-shape guided 3d human avatar generation via diffusion models." arXiv preprint arXiv:2304.00916 (2023).
Jiang, Ruixiang, et al. "AvatarCraft: Transforming Text into Neural Human Avatars with Parameterized Shape and Pose Control." arXiv preprint arXiv:2303.17606 (2023).

- The comparison with AvatarClip is not sufficient. In the supplemental material and Tab 1, results of AvatarClip should be included.

- Since the NeRF in canonical space is pose-dependent, it is theoretically more prone to overfitting. How does the method perform for unseen poses?

- The fact that the shape parameter $\beta$ can vary is not well explained.

- The benefit of the shading and optimization of the SH for environment light is not elaborated sufficiently. What is the albedo before shading? How exactly do you model the irradiance (include rendering equation). The training trick with randomly perturbed SH coefficients is not well motivated. If the goal is to improve disentanglement, why not use some smoothness regularization?

- The animations are shown in a fixed view. It's hard to judge whether loose clothing such as dresses and jackets deform as claimed from a fixed viewing angle.

**Questions:**

It would be great to address my comments above. In particular, I look forward to seeing in the rebuttal:

1. Comparison with AvatarClip

2. Clarification about shading and visualization of the learned albedo

3. Some visual examples of some avatars with loose clothes in walking motion, shown from the frontal view.

4. It would be great to include a user study on the visual quality.


**Limitations:**

Yes

---

> ### Author Rebuttal · Authors · 2023-08-09
>
> We thank the reviewer for the valuable feedback and will address all raised questions.
>
> **Weaknesses**
> - *Concurrent works*: Thanks for the suggestion. We will cite and discuss the suggested references in the final version of the paper. We would like to highlight how our method compares to these 2 approaches.
>   1. AvatarCraft proposes to learn an animatable 3D avatar, but the avatar is very much tied to the underlying SMPL model and does not include instance-specific pose-dependent non-rigid effects. As a result, the geometry of the generated avatars is very close to the underlying body geometry and cannot model loose-fitting clothing, accessories like hats or dresses (see Fig. 5 and 6 of the paper). DreamHuman however is able to handle these cases much more effectively, as shown in our qualitative results.
>   2. DreamAvatar can better capture geometric variations beyond the underlying SMPL body model, but has the major disadvantage that it needs to be retrained every time for a new pose which makes it computationally very expensive. In contrast, our model, once trained, can be easily reposed at no extra cost. Additionally, avatars produced using DreamAvatar exhibit geometric and texture artifacts.
>   3. Overall, by inspecting the quality of the generated avatars, one can see that our method is able to create significantly more realistic avatars, with better texture and geometry, and can be reposed at test time at no major additional cost.
>
> - *AvatarCLIP comparisons*: We did not include comparisons with AvatarCLIP on Table 1 because it was trained using CLIP as supervision, so the CLIP metrics would be biased towards methods that optimize CLIP similarity losses. AvatarCLIP explicitly optimizes the CLIP similarity score. Even if a CLIP model with a different architecture is used for training and testing (e.g. ViT B16 vs ViT B32), all these models were still trained on the same data with the same strategy. Similar observations have been made in Table 1 of DreamFusion in their comparison with the previous work DreamFields. Nevertheless, as requested we run AvatarCLIP using the publicly available implementation on the same 160 prompt set and report the numbers: R-Precision=0.855, Top-3=0.962, Top-5=0.981. We will include these in the final version, and also include additional AvatarCLIP results. We also conducted a user study to assess the visual quality as requested (see below).
>
> - *Pose generalization*: The results shown in the paper (Fig. 1) and in the animations are on unseen poses. From these we can conclude that our model generalizes well on novel poses. We did not observe overfitting because the 3d pose (MoCap) datasets on which the prior was trained are large-scale and diverse.
>
> - *Shape parameter variation*: We optimize the shape parameters as free variables as the shape is part of the identity (in contrast to the pose). The body shape could be constrained by elements of the prompt and we want to make sure that the shape is consistent with those elements in the prompted textual description. For example, the text prompt “a bodybuilder wearing a white shirt” implies that the generated person should have a muscular build. We will add this discussion to the paper.
>
> - *Spherical Harmonics*: For exact details about the Spherical Harmonics model we use please refer to our answer to reviewer KMW9. The primary goal of randomizing SH coefficients is to support the geometry learning (similar to Dreamfusion’s randomized light direction). Randomizing SH coefficients in some cases additionally helps to decouple albedo and lighting, e.g. a dark shirt can be obtained with the absence of light or with dark color. When randomizing the light, the former is no longer possible. However, albedo estimation was not the primary goal and we would like to investigate further in future work. Thank you for the suggestion.
>
> - Animations: We plan to release more high-resolution animation results from varying viewpoints in the final version to better showcase how our method handles clothing deformations. All animations in the supplementary material were rendered from the same viewpoint for consistency reasons. We added an unrolled animation for a skirt from a front view in Figure 7 of the rebuttal, as well as a video animation for a dress in the link (see response to AC as per the conference’s guidelines for submitting videos).
>
> **Questions**
> 1. We added a CLIP-based comparison with AvatarCLIP as well as a user study as mentioned previously in our response.
> 2. We added renderings of the albedo in Figure 5 of the rebuttal PDF. Overall the albedo looks plausible. For the shading please refer to our answer in the previous question.
> 3. We added a rolled-out animation in the rebuttal PDF (Fig. 7)  and also included videos in the response to AC section. We will include more results like this in the final version.
> 4. Following the reviewer's suggestion we conducted a user study to assess the quality of our results. We used 20 text prompts from the AvatarCLIP website from the “General Description” category. We run our method on those 20 text prompts and rendered the final results in the rest pose from the front and the side. We did the same for the precomputed meshes that the AvatarCLIP authors provide in their website. We then asked the users to rate the two methods on (a) the perceived agreement between the renderings and the input text, and (b) the perceived visual quality of the generated avatar. The ratings were on a scale from 1-5 with 1 meaning “very bad” and 5 “very good”. Our method achieved an average rating of 4.45 for “Agreement with text” and 4.16 for “Visual Quality”. AvatarCLIP scored 2.96 on “Agreement with text” and 2.49 on “Visual Quality”. In terms of the scale of the study, the user study we carried out is slightly larger than the one in AvatarCLIP (Ours: 20 images with 25 raters, AvatarCLIP: 8 images with 22 raters). We will include the full list of text prompts used in the Supplementary Material.

---

> > ### Comment · Reviewer_FnWG · 2023-08-18
> > **Satisfied with the answers**
> >
> > Thank you for the explanation.
> > I think the answers to shape parameter variation and optimizing environment light SH is important details, and should be included in the main paper.
> >
> > I appreciate the comparison with concurrent work.
> >
> > Finally, I agree with other reviewers in urging the authors to release their code.
> >
> > My final rating is accept.

---

### Official Review · Reviewer_hMhh · 2023-07-01

**Soundness:** 3 good
**Presentation:** 3 good
**Contribution:** 3 good
**Rating:** 6
**Confidence:** 4

**Summary:**

This paper presents a method to generate animatable 3D human avatars from text. The pipeline is similar to DreamFusion and is built upon the Nerf representation and diffusion model. However, a key difference is that an imGHUM body model is introduced as prior, which allows for the construction of a deformable Nerf representation and a 3D animatable human model. This design not only enables animation capabilities but also effectively addresses the anthropometric consistency issues. In addition, a semantic zooming loss is proposed to refine details in body regions like the face and hands, resulting in a more photo-realistic overall quality.
Quantitative and qualitative comparisons are conducted with state-of-the-art baselines such as DreamFusion and AvatarCLIP. The extensive results demonstrate that the proposed method outperforms previous approaches across all metrics. The visual quality and geometry detail are particularly impressive. Furthermore, the study includes an analysis of different components to highlight their importance within the framework.


**Strengths:**

- The paper is well-written and easy to follow.
- The overall results are impressive, especially the appearance and the geometry details of the generated 3D human avatar.
- Although each component can be seen in previous works like DreamFusion and AvatarCLIP, this work did a good job on putting all losses and modules together properly and achieving promising 3D human avatar modeling.
- Extensive ablation experiments are conducted to show the importance of proposed components.
- The proposed semantic zooming loss is interesting and effective, which largely improves the visual quality and helps to generate sharper, higher-quality textures.


**Weaknesses:**

- It’s not clear to me how to decide the shape parameters for the imGHUM model during optimization. If it is optimized together with the Nerf model, will it introduce additional training costs?
- The overall computation cost is not clearly listed and compared. It would be better to report the optimization time and inference time for a single model/text prompt for the proposed method and baselines.
- It would be better to show more quantitative results for the semantic zooming loss since it's one of the key contributions of this work.


**Questions:**

- Is there any quantitative result for more direct evaluation regarding the view consistency and pose dependency of the generated 3D avatar?



**Limitations:**

- As shown in Figure 7, artifacts can be found in the soldier avatar generated by DreamHuman. The generated avatar has more than two hands, and the method struggles with generating correct accessories. This can be really hard for the current model design since it's built on imGHUM body model.

---

> ### Author Rebuttal · Authors · 2023-08-09
>
> We thank the reviewer for the valuable feedback and will address all raised questions.
>
> **Weaknesses**
> - *Shape parameters*: The shape parameters are treated as additional parameters of the overall model during optimization and we compute the gradient of the loss with respect to them as if they were trainable weights of the neural network. This does not introduce any significant computational cost (only 10 additional latent shape parameters), whereas the NeRF MLPs have over 1 million parameters.
> - *Computation cost*: Training our method for 50K iterations takes around 6 hours per example on a TPU v4 machine. After training we can render the subject in an arbitrary pose at 512x512 resolution in 2.6 seconds, and at 256x256 in 0.67 seconds. The rendering time is similar with DreamFusion, however DreamFusion trains only for 15K iterations which takes about 1.5 hours. We need a larger number of iterations until convergence because we train a dynamic and not a static avatar. AvatarCLIP takes up to 10 hours to train on a single GPU with 32GB of memory. These are all typical times for training NeRF-based architectures.
> - *Semantic zoom*: We would like to thank the reviewer for their suggestion. We will add more results in the supplementary material.
>
> **Questions**
> - To the best of our knowledge there are no quantitative metrics to assess the view consistency and the quality of the pose-dependent deformations. To evaluate the quality of text-based 3D generations most methods rely on imperfect 2D metrics, such as the CLIP R-Precision score. However these are not very good for measuring the quality of the 3D geometry or the view-consistency of the generations, as they only measure the agreement of the rendering with the text prompt.
> We train a NeRF with no view-dependent effects, so our model is by design viewpoint consistent. The pose consistency relies on pose-sensitive deformations learned from data.
>
> **Limitations**
> - Indeed this is an interesting failure case of our method. In fact after inspecting the result we found that although the extra hand is painted on the military uniform, no extra geometry is created. In practice we observed that text-to-image models tend to exhibit certain pose biases, i.e. when prompted to generate an image of a soldier, most generated images include soldiers holding a weapon. In the vast majority of cases, randomization during training helps mitigate this issue, but in rare cases errors like this may occur. In fact in Figure 6 of the rebuttal PDF we show that if we run again the optimization with a different random seed we get a correct result this time.

---

> ### Comment · Reviewer_hMhh · 2023-08-16
> **Thanks for your response.**
>
> Thank you for the detailed feedback and the new qualitative results, which address my earlier concerns. I believe this is solid work with several innovations. And I agree that making the model accessible for the purpose of reproducibility would serve as valuable assets for the broader community.

---

### Official Review · Reviewer_urhH · 2023-07-05

**Soundness:** 4 excellent
**Presentation:** 3 good
**Contribution:** 3 good
**Rating:** 8
**Confidence:** 5

**Summary:**

This work proposes a method for text-driven human avatar generation. It combines animatable human nerf and diffusion model to implement avatar generation and animation. This work produces photorealistic avatars with high-quality details by incorporating spherical harmonics lighting model and semantic zoom. Extensive experiments demonstrate SOTA performance and the effectiveness of each design in the proposed framework.

**Strengths:**

1. This is the first diffusion-based work that successfully produces photorealistic animatable 3D human avatars.
2. This work shows temporally consistent animation results. I believe this can open up more application possibilities for optimization-based avatar generation methods.
3. The incorporation of spherical harmonics lighting model can alleviate the long-standing issue of unrealistic over-saturated color for text-driven 3D object generation.
4. The semantic zoom loss is simple yet effective to improve the quality for detail regions such as face, arm, and hand.


**Weaknesses:**

1. The imGHUM is designed for the whole body, it contains parameters for hand and facial expressions. But there is no result for the animation of facial expressions and hand poses. It could be better if the authors can provide animation results to show the controllability of these details. I think this will also help prove the necessity of semantic zoom loss.
2. I believe this work uses a more powerful diffusion model, and imGHUM is not completely open source. These two points will limit access to the proposed model. Is there any plan to release an online demo or interface for users?
3. Although the authors mentioned the training strategy in Sup Mat, there is no clear description of the computation cost of the proposed method.

**Questions:**

1. How to determine the rendering camera poses for each part in the semantic zoom?
2. Which diffusion model is used in this work? Is this diffusion model finetuned on human body images?


**Limitations:**

Yes, addressed.

---

> ### Author Rebuttal · Authors · 2023-08-09
>
> We thank the reviewer for the valuable feedback and will address all raised questions.
>
> **Weaknesses**
> 1. *Hands and facial expressions*: Thanks for the very good suggestion. For the rebuttal we have produced examples of renderings with varying facial expressions (Figure 2) and hand poses (Figure 3) and we will include them in the final version. Note that this is similar to varying shape coefficients (Figure 1 of the Supplementary Material), as we did not train with varying hand poses or expressions. Nevertheless, our model is able to generalize by leveraging the capabilities of imGHUM.
> 2. *Different diffusion models*: We use the same model as DreamFusion to ensure a fair comparison. Notice however, that our model is not necessarily more powerful than other public domain models like e.g Stable Diffusion. We included some example results by replacing Imagen with Stable Diffusion in Figure 4 of the rebuttal PDF. The performance of the 2 variants is perceptually very similar, thus showing that our model is agnostic to the architecture of the underlying text-to-image diffusion model. Regarding the availability of imGHUM, it is distributed under an academic license. We will strive to make a demo available.
> 3. *Computation cost*: Our model is trained for 50K iterations and this takes about 6 hours on a TPU v4 machine, which is similar to other competing methods (1.5 hours for DreamFusion for static avatars,, 4-10 hours for AvatarCLIP). After training we can repose and render our model very fast, and takes 2.6 seconds for rendering a 512x512 image, and 0.67 seconds for rendering a 256x256 image.
>
> **Questions**
> 1. *Semantic zoom*: Given a particular body pose and shape we can get the 3D joint locations for the body part. Given these we place the camera at a reasonable distance from the body part to minimize perspective distortion effects. We also compute a range of focal lengths that ensures that the body part occupies a large enough portion of the image.
> 2. *Diffusion model*: We use Imagen, a general-purpose text-to-image diffusion model. We did not finetune on human body images. As we show in Figure 4 of the rebuttal PDF, our method works with other diffusion models such as the open-source Stable Diffusion.

---

> > ### Comment · Reviewer_urhH · 2023-08-18
> >
> > Thank the authors for the answers. The results in the rebuttal file are nice and persuasive. All of my concerns have been addressed, and I will keep my rating.

---

### Official Review · Reviewer_KMW9 · 2023-07-09

**Soundness:** 3 good
**Presentation:** 3 good
**Contribution:** 3 good
**Rating:** 6
**Confidence:** 3

**Summary:**

This paper proposes a method to generate high-quality and animatable 3D human from textual input.

**Strengths:**

- The result is good, showing clear improvement compared to the previous text-to-3D method.

- The method can be learned without 3D GT.

- The ablation study is thoroughly done.

**Weaknesses:**

- What is the run-time at inference?

- Density loss: I think this can work when the suject is wearing a tight clothing. However, wouldn't this confuse the network when the subject is wearing a clothing with large deformatation (i.e., more gap is present between the body model and the actual geometry)?

- What is w_i of equation 5?

- What is the proposal weights L_p in L195?

- It is written that 4 renderings are supervised for a single training step (supplementary material L7). Does this mean that 4 randomly selected semantic parts (zoomed-in semantic parts) of the same pose are rendered? Or does this mean that single semantic part is rendered from different camera pose and body pose? Also, would rendering and supervising predictions less than four (e.g., when using single GPU with less memory) degrade the performance? Discussion on the running environment and performance would be helpful for the readers.

- Lack of implementation details, making it difficult to reproduce. Also, no plan on supporting the reproducibility has been presented.

**Questions:**

- What is the rendering resolution at inference time?

- How are the query locations sampled? Are they sampled in free space or bounded box?

- Which method was used to perform rendering with spherical harmonics lighting model?

- How are the camera parameters (both extrinsic and intrinsic) set? Since there are no known camera parameters for the images generated with Diffusion models, I am wondering how the authors set the parameters to render images.

- Is there a reason behind choosing mip-NeRF360 as the backbone? Would using basic NeRF degrade the performance?

- What is s (the output of imGHUM) exactly? Is it an index of the nearest vertex on the body?

- Although the lighting result is much better than the previous work (Dream Fusion), still it looks unnatural. What is the reason behind this and how it can be improved?

**Limitations:**

- The idea is interesting and the results are great. However, there is a concern regarding the reproducibility.

---

> ### Author Rebuttal · Authors · 2023-08-09
>
> We thank the reviewer for the valuable feedback and will address all raised questions.
>
> **Weaknesses**
> - *Runtime*: The rendering time for a 512x512 image is 2.6 seconds, whereas for 256x256 it is 0.67 seconds. These timings were measured on a TPU v3 chip with 8 cores.
> - *Density loss*: The purpose of this loss is to act as a regularizer and encourage the model to preserve important details such as facial features as we show in Fig. 4 of the Supplementary Material. Our results in Figure 2 of the main paper show that we can handle loose clothing such as dresses, something that previous work like AvatarCLIP cannot.
> - W_i are the weights of the samples along a ray as defined in Eq (1). The formulation we use is the same as in Eq (10) of Ref-NeRF [56].
> - *Proposal weights*: We use mip-NeRF 360 as our NeRF backbone, which consists of 2 MLPs, the proposal MLP and the NeRF MLP. The proposal weights are the rendering weights as defined in Eq (1) but for the proposal MLP. L_p as proposed in mip-NeRF 360 is used to supervise the proposal MLP and penalizes the proposal weights when they underestimate the distribution of the NeRF MLP. Please refer to Section 3 of mip-NeRF 360 for a more detailed discussion.
> - *Training details and environment*:
> For each view we randomly select camera poses, semantic parts and body poses. The selection probabilities for the semantic parts are listed in Table 1 of the Supplementary Material. We combine each view with a different semantic part and a different pose. We include a discussion of the running environment in Section 1 of the Supplementary Material. Our network can be trained on TPUs or GPUs and with 16GB of memory we can fit 1 view per device.
> The model still has acceptable performance when trained with 2 views per step but it needs more iterations to converge. With a smaller number of views, we noticed that it’s harder for our network to learn the dependency of f on $\theta$ (Eq 3), probably because of the higher variance in the gradients, similarly to training a neural network with a small batch size. With 1 view and all other hyperparameters unmodified the  optimization becomes unstable. In Figure 1 of the rebuttal PDF we show examples trained with 2 and 4 views.
> - *Implementation details*: Thank you for pointing this out. We have included the hyperparameters and training strategy in the main paper and the supplementary material. We will expand this section in the supplementary material and include a more detailed algorithm section. As we show in Figure 4 of the rebuttal PDF, our method is generic enough and works with other open-source diffusion models such as Stable Diffusion.
>
> **Questions**
> - *Render resolution*: Since our model is NeRF-based and we trained it with camera randomizations, we are not tied to a particular resolution. In the paper figures we used a resolution of 512x512 pixels. In the Supplementary Material videos we used 256x256 due to document size constraints.
> - *Query location sampling*: We assume that the scene resides inside a unit sphere centered around the origin and we use an additional scaling factor to make sure the human fits in the unit sphere.
> - *SH*: We use a simple Spherical Harmonics Diffuse Lighting model. Given the 9 light source coefficients $c_j$, for each pixel in the image we compute the dot product between the 9-element vector of the learnable light coefficients and the vector of the corresponding spherical harmonics basis functions $h_j$ and then we use this to compute the final shaded color given the albedo. More specifically, in our formulation, the shaded pixel value $s_i$ at pixel $i$ with albedo $\alpha_i$ and unit surface normal $(x_i, y_i, z_i)$ is $s_i = \alpha_i \cdot \left(\sum_{j=0}^8 c_j h_j(x_i, y_i, z_i)\right)$. More details can be found in Spherical Harmonics Lighting: The Gritty Details (https://3dvar.com/Green2003Spherical.pdf). We will include all these details in the Supplementary Material.
> - *Camera parameters*: We use a reasonable distance from the subject to ensure that we don’t have severe perspective distortion effects. For example, for the full body shots the camera distance is between 2.5 and 4 meters away from the subject and the camera zoom is chosen such that the person takes up a large portion of the image and is fully visible in most frames.
> - *NeRF backbone*: mip-NeRF 360 has been shown to achieve better results than NeRF and is computationally cheaper because it needs fewer samples per ray than the standard NeRF. However any other competitive Nerf method can be used instead.
> - *imGHUM*: For a given pose θ and a point (x,y,z) in the 3D space, imGHUM returns the distance d of the point from the surface along with a semantic code s that associates it with the closest point on the surface of the posed GHUM model. If e.g. the closest point on the mesh is the tip of the index finger, then s is the 3D position of the tip of the index finger in the GHUM template mesh. The semantic code s is not an explicit vertex id, but rather a continuous surface mapping. We kindly refer the to the imGHUM paper for a more detailed explanation.
> - *Color quality*: It has been reported in previous works (e.g. DreamFusion) that the use of the SDS loss tends to produce saturated colors, and we have observed similar effects. Besides our proposed spherical harmonics estimation strategy, one way to improve the appearance could be to augment the training with an additional loss that attempts to match the statistics of our rendered images to those of natural images.

---

> > ### Comment · Reviewer_KMW9 · 2023-08-18
> > **Response to the author rebuttal**
> >
> > I appreciate the authors for their time and efforts put into the rebuttal. The author rebuttal has successfully addressed my concerns. I will keep my rating.

---

### Author Rebuttal · Authors · 2023-08-09

We would like to thank the reviewers for their valuable feedback.

- Reviewer KMW9 states that our method shows clear improvements over previous methods without the need for 3D ground truth data, and that we have a comprehensive ablation study.
- Reviewer urhH notes that our method produces animatable photorealistic 3D avatars, with temporally consistent animation results, and appreciates our novel components that improve the overall reconstruction quality.
- Reviewer hMhh highlights that our 3D generation results are impressive, appreciates our extensive ablation study and notes that the novel components of our method contribute in achieving higher-quality results.
- Reviewer FnWG praises the substantially better visual quality compared to previous methods and the importance of our semantic prompting.

We will address the questions raised by each reviewer separately under each review. The summary of the rebuttal is as follows:
- We performed extra evaluations against AvatarCLIP and conducted a user study on the visual quality of the results.
- We demonstrated that our method also works with open-source diffusion models such as StableDiffusion, thus addressing potential concerns about the reproducibility.
- We added extra clarifying qualitative results.
We will include all these results in the final version of the paper.

---

### Decision · Program_Chairs · 2023-09-21

**Decision:**

Accept (spotlight)

**Comment:**

This paper aims at the task of generating an animatable human avatar given a text prompt as input. The technical contributions include: a) building on top of an implicit 3D human pose and shape model (i.e., imGHUM) which provides not only geometry but also deformation of each query point. b) adopting a semantic zoom training strategy that enhances details on small body parts (e.g., face and hands) and c) using a physics-based shading formulation that produces more realistic texture. Experimental results and ablation studies demonstrate the effectiveness of the proposed modules and its superior performance against existing methods (e.g., DreamFusion, AvatarCLIP).

The strengths of the paper are:
- All reviewers agree that the results of the paper are impressive, including photo-realistic synthesis, detailed geometry and temporally consistent animation.
- Thorough ablation studies to verify the contribution of proposed modules (Reviewer hMhh, KMW9)
- The semantic zoom training strategy and physics-based shading module are both simple yet effective (Reviewer urhH, hMhh, FnWG)

Pre-rebuttal, the reviewers mentioned the following common concerns with other clarification questions:
- Inference time. (Reviewer KMW9, urhH, hMhh)
- Reproductivity. (Reviewer KMW9)
- Missing comparisons and ablation studies. (Reviewer FnWG)

After rebuttal, reviewers acknowledged that all concerns are addressed except reproductivity. After reading the paper, reviews and rebuttals, I agree with the reviewers that this work provides valuable technique and insight to the emerging text-to-3D community, especially its significant performance compared to existing works. Thus I recommend for acceptance. Meanwhile, I urge the authors to release their code to facilitate future research.